# Clinical Insights into the Treatment of Patellofemoral Instability with Medial Patellofemoral Ligament Reconstruction: Pearls and Pitfalls—Lessons Learned from 20 Years

**DOI:** 10.3390/jpm13081240

**Published:** 2023-08-09

**Authors:** Kata Papp, Bernhard M. Speth, Carlo Camathias

**Affiliations:** 1Department of Traumatology and Orthopaedic Surgery, Kantonsspital Aarau, CH-5000 Aarau, Switzerland; 2Department of Orthopaedic Surgery, University Children’s Hospital Basel, CH-4031 Basel, Switzerland; 3Orthopädie für Kinder & Jugendliche, Praxis Zeppelin, CH-9016 St. Gallen, Switzerland; 4Faculty of Medicine, University of Basel, CH-4031 Basel, Switzerland

**Keywords:** medial patellofemoral ligament reconstruction, MPFL, femoropatellar instability, young athletes

## Abstract

Patellofemoral instability is a prevalent cause of pain and disability in young individuals engaged in athletic activities. Adolescents face a particularly notable risk of patellar dislocation, which can be attributed to rapid skeletal growth, changes in q-angle, ligamentous laxity, higher activity levels, and increased exposure to risk. Specific sports activities carry an elevated risk of patellar dislocation. Younger age and trochlear dysplasia present the highest risk factors for recurrent patellar dislocations. International guidelines recommend conservative therapy following a single patellar dislocation without osteochondral lesions but suggest surgical intervention in recurrent cases. In this study, we have compiled current scientific data on therapy recommendations, focusing on MPFL (medial patellofemoral ligament) reconstruction. We discuss patient selection, surgical indications, graft selection, location and choice of fixation, graft tensioning, and postoperative care.

## 1. Introduction

Patellofemoral instability is a prevalent cause of pain and disability in young individuals engaged in athletic activities. It often leads to patellar dislocations, which are more common among adolescent girls aged 10 to 17 [1]. However, recent studies indicate higher injury rates among boys, possibly due to their participation in sports such as football and wrestling [2]. Nevertheless, when comparing both genders in sports where males and females participate equally, females appear to have a higher risk of patellar dislocations. The risk of such injuries increases not only with athletic activity but also with military service before the age of 20 [3]. Certain sports, including girls’ gymnastics and soccer, as well as boys’ football and wrestling, carry an elevated risk of patellar dislocation [2].

According to a cohort study by Sanders et al., the overall incidence of patellar dislocation is 23.2 per 100,000 person-years [4]. This study also revealed a significantly higher incidence of patellar dislocation among adolescents than previously estimated. Therefore, the risk of patellar dislocation in this age group is notable, attributed to rapid skeletal growth, changes in the q-angle, ligamentous laxity, higher activity levels, and increased exposure to risk.

Typically, dislocations occur through a mechanism involving knee flexion and rotation, usually as a noncontact but traumatic event [5]. Contact with other players appears to increase the risk of dislocation in males, while females tend to experience femoropatellar instability without direct player contact. Mitchell et al. explain that anatomic factors play a more prominent role in females with patellar dislocations, including a higher prevalence of ligamentous laxity and changes in the q-angle, among others [2].

Various risk factors contribute to patellar dislocations, such as trochlear dysplasia, patella alta, malalignment syndromes, axial leg deformities, and neuromuscular disorders. Among recurrent patellar dislocations, younger age and trochlear dysplasia are the highest risk factors [6]. The medial patellofemoral ligament (MPFL) is injured in approximately 87% of dislocation cases. Alongside the bone structure (static stabilizer) and the surrounding muscles (dynamic stabilizers), the MPFL is the primary passive restraint against lateralization in the initial 30° of knee flexion [7,8]. Rupture of the ligament typically occurs when it elongates to around 26 mm under a tensile force of 208 N [9].

International guidelines suggest conservative therapy after a single patellar dislocation without osteochondral lesions but recommend treatment in cases of recurrence. MPFL reconstruction is a less invasive procedure than most bony interventions, as it leaves the cartilage untouched. A systematic review of Testa et al. comparing MPFL reconstruction and trochleoplasty demonstrates significant improvement in postoperative clinical scores for both procedures [10]. Hurley et al. indicate that reconstructing the MPFL reduces rates of recurrent instability and re-dislocation postoperatively, compared to MPFL repair or nonoperative management [11]. Previous research by the same authors suggests that MPFL reconstruction can be highly effective for patients with patellofemoral instability and no or moderate risk factors [12].

Although MPFL reconstruction may require less effort than bony procedures, postoperative rehabilitation is complex and lengthy. Recurrent instability after MPFL reconstruction is a common issue [13]. Therefore, it is essential for physicians to select the most suitable operative method with minimal effort and negligible risk for patients.

The purpose of this article is to review the current literature, analyze the risks and benefits of single MPFL reconstruction in athletic youth with patellofemoral instability, and outline the key aspects of this procedure. Our aim is to provide physicians with a practical guideline for planning therapy in young individuals, as utilized in our therapy consensus.

## 2. Materials and Methods

We conducted a search in PubMed for recent literature that describes the outcomes of MPFL reconstruction between 2000 and 2022. In order to provide a comprehensive analysis, we also included two studies from before 2000 due to their relevance [7,14]. We did not impose any specific inclusion criteria regarding article type or level of evidence. However, we did exclude articles that focused on operations involving concurrent or previous procedures, as well as those involving patients with multi-ligamentous or osteochondral injuries.

## 3. Results

In recent years, both the number of publications on this topic and the frequency of this operation worldwide have increased. In this article, we aim to present the most significant findings regarding solitary MPFL reconstruction.

### 3.1. Patient Selection

Careful patient selection is one of the most critical factors contributing to postoperative success, as it helps determine the correct indication for surgery. In the case of patients with lateral patellar instability and recurrent dislocations without patellofemoral arthrosis, the primary selection is for MPFL reconstruction.

It is crucial to analyze the bony anatomy of each individual and plan any necessary concomitant operations. Failure to address trochlear dysplasia and neglecting the elevated TTTG (tibial tuberosity to trochlear groove) distance are the most common causes of postoperative failure [15]. Therefore, it is essential to take these factors into account. A higher degree of trochlear dysplasia, patella alta, genu valgum, and elevated femoral antetorsion are indications for bony procedures. In such cases, a bony procedure would be the preferred option, as it can effectively address these specific anatomical abnormalities and provide better outcomes.

The ideal patient for MPFL reconstruction should be younger than 30 years and free from obesity or cartilage damage. Additionally, it is worth noting that women tend to have poorer clinical outcomes [16]. According to a review by Migliorini, younger individuals who engage in athletic activities are more likely to experience re-dislocations after MPFL reconstruction [17].

### 3.2. Surgical Indications

Patellar instability during daily activities or sports, which has not responded to nonoperative therapy, indicates the need for operative treatment. An MRI can help confirm the diagnosis by showing a bone bruise at the typical lateral patellar facet and medial side of the lateral femoral condyle, while also ruling out any accompanying injuries [18]. In certain cases, when the MRI quality is good, it may also be possible to identify the site of the rupture.

### 3.3. Graft Selection

Similar to the anterior cruciate ligament (ACL), the medial patellofemoral ligament (MPFL) can be reconstructed using various options such as the quadriceps tendon, adductor magnus tendon, hamstrings, or synthetic materials. However, there is currently no consensus on the optimal graft choice [19]. Some evidence suggests that the semitendinosus tendon may yield better outcomes compared to the gracilis graft, as reported in a systematic review by Migliorini et al. [20].

A straightforward technique described by Steensen offers a convenient method for utilizing the quadriceps graft without the need for drilling a patellar tunnel, thus reducing the risks of patellar fracture or cartilage damage [21]. This technique proves beneficial for skeletally immature patients as it avoids tunneling near the growth plate, a practice we also employ in our procedures.

Achieving an appropriate graft length is crucial for a successful reconstruction, as it can impact the decision between double- or single-bundle techniques.

### 3.4. Location of Fixation

Another crucial factor for achieving a favorable outcome is the positioning of the tunnel and the site of fixation. The femoral attachment is considered the most critical point that affects isometry and ensures proper functioning [21]. The anatomic femoral insertion point lies between the adductor tubercle and the medial femoral epicondyle. Placing the tunnel too anteriorly and proximally is a common mistake, as it results in excessive pressure on the medial side [22].

However, it is important to note that the ligament exists as a complex with a broad attachment. Approximately 57% of the insertion is located at the patella, while 43% is found at the deep quadriceps tendon [23,24]. Tanaka also described the midpoint of this ligament complex as the point on the proximal cartilage border of the patella where the quadriceps tendon attaches. In young individuals, the ligament’s origin is typically located 4.7 to 10 mm distally from the physis of the distal femur. However, this distance can vary with growth depending on the age of the patient [25]. Patients with an open physis present a special group with unique surgical challenges. The target isometric fixation point for an MPFL graft is situated just anterior to the posterior cortex of the femur and proximal to the Blumensaat line.

Regarding the patellar tunnel, it should be placed at the anatomical insertion site on the proximal two-thirds of the medial side of the patella, depending on the graft choice and technique [21]. Positioning the tunnel too anteriorly can lead to a fracture, while placing it too posteriorly can cause articular damage [26]. Therefore, obtaining an accurate lateral fluoroscopy view of the knee during the procedure is essential to determine the correct tunnel sites [27]. It is recommended to drill 3.2 mm tunnels instead of 4.5 mm to reduce the risk of fracture [28]. 

Techniques that are suitable for adults may not be appropriate for younger individuals due to the risk of injuring the physis and causing growth disturbances. The MPFL is closely related to the physis, making it susceptible to injury during temporary hemiepiphyseodesis of the distal femur [29]. Therefore, when choosing the location of fixation, it is important to select a distal point from the physis for a more isometric outcome compared to a proximal placement of the femoral insertion [30]. Special care should be taken with the MPFL in cases where simultaneous temporary hemiepiphyseodesis and MPFL reconstruction are planned, especially when aiming to correct the leg axis by implanting special plates on the medial femur for growth control.

### 3.5. Choice of Fixation

There are several methods available to address graft fixation, ranging from traditional screws to sutures. The two primary techniques for fixation are suture-based and tunnel-based approaches [31]. According to Shah et al., it is challenging to determine which approach is superior in terms of complications [31]. However, selecting a method that involves through-tunnel fixation in the femoral region is considered crucial for achieving the strongest reconstruction [9].

When it comes to fixing the patellar end, it is important to ensure minimal prominence to prevent implant-related pain and the potential need for subsequent removal. Implant pain is a common issue at the femoral insertion site and, in some cases, may require the removal of screws [26].

### 3.6. Graft Tensioning

Excessive tightening of the graft during surgery may lead to medial dislocations of the patella or a loss of knee motion, which could necessitate further operations [26].

In severe and uncommon cases, this restricted motion can result in a condition known as arthrofibrosis. Overconstraining the graft increases the pressure on the patellofemoral joint, causing pain and potentially leading to the development of arthrosis over time [32]. Conversely, insufficient constraint of the graft can result in recurrent dislocations.

It is crucial to consider the type of graft selected for the reconstruction. Hamstrings, for instance, tend to be stiffer and more durable than the original MPFL, requiring special care to avoid overtightening [14].

### 3.7. Postoperative Care

The postoperative care after MPFL reconstruction has not been standardized yet. However, it typically involves limited weight-bearing for several weeks and a gradual increase in the range of motion over a period of six weeks.

In a recent retrospective study conducted by Gulbrandsen et al., it was observed that there was a significant reduction in postoperative pain compared to the preoperative scores [33].

This finding remained consistent regardless of whether patients returned to their preoperative activity level or not [34]. It is worth noting that around 91% of patients were able to resume sports activities following isolated MPFL reconstruction. Moreover, within the same study, 67% of patients returned to the same or even higher level of activity than before their operation, with an average time of 10 months.

The reasons for this can be attributed to the fear of re-injury as well as limited knee flexion and extension strength of the operated knee. However, it should be acknowledged that the exact reasons cannot be definitively determined in every case [35].

Postoperative rehabilitation plays a crucial role in the successful return to sports activities. It is strongly recommended to follow a formal physical therapy program rather than relying solely on YouTube videos, as these videos tend to score low in terms of quality, reliability, understandability, and actionability [36].

## 4. Discussion

The stabilizing role of the medial patellofemoral ligament (MPFL) reaches its peak at 25–30° of flexion, showing the maximum change in strain. However, beyond 60° of flexion, the trochlea becomes the primary stabilizer [37].

Depending on the anatomical configuration, there are numerous possibilities to address the main issue of patellar instability. The key is to identify the precise factors causing instability, which often include not only a ligamentous deficiency but also skeletal anomalies such as trochlear dysplasia or patella alta, femoral/tibial rotational problems, or leg axis deformities (e.g., genu valgum). It is important to avoid relying solely on MPFL reconstruction in the presence of these anatomical changes.

Isolated MPFL reconstruction has shown reasonable patient satisfaction, reduced pain scores, and a successful return to sports on average, provided that patient selection and indications are carefully considered [38]. In carefully selected patients undergoing isolated MPFL reconstruction, the rate of recurrent instability is low, with an approximate risk of 1.9% [35]. Similar findings regarding dislocation rates were observed in a systematic review conducted by Testa et al., which compared MPFL reconstruction to trochleaplasty. Both groups showed similar dislocation rates of approximately 2% [10]. However, the review also indicated that while the patellofemoral joint was mostly stable postoperatively, most of the investigated outcome parameters did not return to normal values [10]. When discussing the operation’s indication with athletic patients, it is important to emphasize that they are likely to return to their sports activities, albeit at a lower intensity level than before the operation [35].

Even the possibility of conservative treatment should be discussed with patients, particularly after a single dislocation without significant variations in the bony anatomy. Adolescents commonly experience a tear of the MPFL at the insertion site, whereas adults often have mid-substance ruptures. Vavken et al., in their systematic review, found no significant difference in clinical scores and recurrence rates between nonoperative and surgical treatments of patellar instability in children and adolescents [39].

## 5. Conclusions

Based on our literature research and personal experience, we recommend considering the following aspects to define the most accurate therapy:For isolated MPFL reconstruction, it is preferable to consider individuals under the age of 30 without arthrosis or obesity.MPFL reconstruction alone is not suitable for patients with bony anomalies such as trochlear dysplasia, increased femoral antetorsion, genu valgum, or patella alta. In such cases, bony correction should be considered to achieve successful postoperative outcomes.There is no evidence supporting the necessity of surgical therapy after a single patellar dislocation. However, the psychological stress and impact on everyday activities and sports should be taken into account.When harvesting the graft, it is important to ensure a sufficient length of approximately 7 cm.The femoral fixation of the graft should be placed between the adductor tubercle and the medial femoral epicondyle. Intraoperative imaging from a pure lateral view can assist in locating the correct tunnel placement.In patients with open physis, the location of femoral fixation should be distal to the physis to ensure optimal functionality without causing growth disturbances.The patellar fixation should be positioned on the proximal medial two-thirds, and a 3.2 mm drill head should be used instead of 4.5 mm to minimize complications.It is important to avoid prominent hardware in the fixation process.Intraoperatively, the patellar motion should be assessed, and a translation of two or three quadrants should be allowed for, similar to the healthy contralateral side. This helps ensure proper patellar tracking. Additionally, graft fixation should be performed at approximately 30° flexion of the knee, as supported by literature references [28].Postoperative care should include a tailored physical therapy program that addresses the specific needs of the patient. It is important to have open discussions with patients about their expectations regarding returning to sports activities, ensuring that expectations are realistic and aligned with their individual recovery progress.

## Data Availability

Not applicable.

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
