# Peer review of "Clinical Insights into the Treatment of Patellofemoral Instability with Medial Patellofemoral Ligament Reconstruction: Pearls and Pitfalls—Lessons Learned from 20 Years"

_jpm, 2023, doi:10.3390/jpm13081240_

Round 1

Reviewer 1 Report

The topic of this study is very interesting yet very well described in the literature so far. The authors deploy a very comprehensive  pre-intra-post op plan for MPFL reconstruction. However, I see no review here, even in the critical or comprehensive way. The authors make no refer to the number or quality of the papers they found. Instead,they have used a few systematic reviews to make their point. A typical flow chart should be included. 

English language review is needed. Please change operational method/therapy to operative  in lines 69 and 96. When using plural such as several studies then the references should be more than one! Expresions such as "as the cited study clarifies" should be avoided- use the authors name et al. 

At the location of fixation section,for the femoral side please refer to Schottle's point as this is the main landmark (it is described but not named),add literature for that as well

Discussion should be at least one paragraph longer. A comprehensive review turn of the paper would help to that.

Author Response

Thank you for your critics and help. Indeed, the article is no comprehensive review and we wanted to change its category to "opinion". Thank you for your critics regarding the language. Schottle´s point is now included. 

Reviewer 2 Report

Dear Authors,

I read Your paper with interest and appreciation. Below, some remarks I hope will improve the manuscript.

1.     Authors should present precise inclusion and exclusion criteria of evaluated papers.

2.     Authors should present: range of years they searched literature in PubMed, number of entire evaluated papers, number of included and excluded publications.

Author Response

Dear Reviewer, 

thank you for your critics. 

We precised our inclusion and exclusion criteria, although we wanted to change the type of article from "review" to "opinion" before. It will better fit that category.

Reviewer 3 Report

This is a review article analyzing the risks and benefits of the single medial patellofemoral ligament reconstruction in athletic youth in patellofemoral instability cases and determining the cornerstones of this procedure. Although the review deals with an important topic for clinicians, this reviewer has some minor comments/suggestions that the authors need to address before resubmission. Please find them below.

-          Introduction – to contextualize this review, the authors may need to provide more context and background information about the risks and benefits of the single medial patellofemoral ligament reconstruction in athletic youth with patellofemoral instability.

-          It's unclear what the study's relevance is. Authors should note any gaps in the body of knowledge and underline the significance of filling such gaps. Additionally, a stronger justification would lead to a more explicit declaration of the potential implications for practice, which is currently insufficient.

-          This review design may be evidence-based, but often does not meet important criteria to help mitigate bias – frequently it lacks explicit criteria for article selection, and frequently there is no evaluation of selected articles for validity. This should be addressed in the study limitations.

-           A detailed description of the study design is imperatively needed.

-          How many articles have been reviewed in order to arrive at the current conclusion? Was that number of studies enough to provide transferable evidence about the risks and benefits of the single medial patellofemoral ligament reconstruction in athletic youth with patellofemoral instability?

Author Response

Thank you for your detailed critics. 

We wanted to change the article type from "review" to "opinion" beforehead, because it would fit better. The aim was to give a detailed analyze of the technique which can easily be used by clinicians. 

Round 2

Reviewer 1 Report

Thank you for following the suggestions in my review. Indeed the paper reeds better now.  Few spelling errors need to be corrected.

Line 36 change elevated with higher

Line 45  High energy contact sports is the appropriate term

Line 49 Delete etc -either continue with further factors or put a full stop

Line 62 the correct term is trocheoplasty

Line 98 delete elevated and use a high TT-TG...